# N-type organic thermoelectrics: demonstration of $ZT > 0.3$

Jian Liu [1✉], Bas van der Zee[1], Riccardo Alessandri [1,2], Selim Sami [1,3], Jingjin Dong[1], Mohamad I. Nugraha[4], Alex J. Barker[5], Sylvia Rousseva [1,3], Li Qiu [1,3,7], Xinkai Qiu [1,3], Nathalie Klasen[1,3], Ryan C. Chiechi [1,3], Derya Baran [4], Mario Caironi [5], Thomas D. Anthopoulos [4], Giuseppe Portale [1], Remco W. A. Havenith [1,3,6], Siewert J. Marrink [1,2], Jan C. Hummelen [1,3] & L. Jan Anton Koster [1✉]

The 'phonon-glass electron-crystal' concept has triggered most of the progress that has been achieved in inorganic thermoelectrics in the past two decades. Organic thermoelectric materials, unlike their inorganic counterparts, exhibit molecular diversity, flexible mechanical properties and easy fabrication, and are mostly 'phonon glasses'. However, the thermoelectric performances of these organic materials are largely limited by low molecular order and they are therefore far from being 'electron crystals'. Here, we report a molecularly n-doped fullerene derivative with meticulous design of the side chain that approaches an organic 'PGEC' thermoelectric material. This thermoelectric material exhibits an excellent electrical conductivity of >10 S cm$^{-1}$ and an ultralow thermal conductivity of <0.1 Wm$^{-1}$K$^{-1}$, leading to the best figure of merit $ZT = 0.34$ (at 120 °C) among all reported single-host n-type organic thermoelectric materials. The key factor to achieving the record performance is to use 'arm-shaped' double-triethylene-glycol-type side chains, which not only offer excellent doping efficiency (~60%) but also induce a disorder-to-order transition upon thermal annealing. This study illustrates the vast potential of organic semiconductors as thermoelectric materials.

[1] Zernike Institute for Advanced Materials, University of Groningen, Nijenborgh 4, 9747 AG Groningen, The Netherlands. [2] Groningen Biomolecular Sciences and Biotechnology Institute, University of Groningen, Nijenborgh 7, Groningen NL-9747 AG, The Netherlands. [3] Stratingh Institute for Chemistry, University of Groningen, Nijenborgh 4, 9747 AG Groningen, The Netherlands. [4] King Abdullah University of Science and Technology (KAUST), Physical Sciences and Engineering Division (PSE), KAUST Solar Center (KSC), Thuwal 23955-6900, Saudi Arabia. [5] Center for Nano Science and Technology @PoliMi, Istituto Italiano di Tecnologia, via Pascoli 70/3, 20133 Milano, MI, Italy. [6] Ghent Quantum Chemistry Group, Department of Inorganic and Physical Chemistry, Ghent University, Krijgslaan 281 (S3), B-9000 Gent, Belgium. [7]Present address: Yunnan Key Laboratory for Micro/Nano Materials & Technology, National Center for International Research on Photoelectric and Energy Materials, School of Materials and Energy, Yunnan University, Kunming 650091, PR China. ✉email: Jian.liu@rug.nl; l.j.a.koster@rug.nl

Thermoelectric (TE) materials can be used as a solid-state and green energy technology for converting waste heat into electricity or directly using electrical power for cooling and heating[1,2]. The thermoelectric performance is defined by the figure of merit $ZT = S^2\sigma T/\kappa$, where $S$, $\sigma$, $T$, and $\kappa$ represent the Seebeck coefficient, the electrical conductivity, the absolute temperature, and the thermal conductivity, respectively[1,3]. Glen Slack proposed that an ideal TE material should be a "phonon-glass electron-crystal" (PGEC), which "means a material in which the phonon mean free paths are as short as possible and in which the electron mean free paths are as long as possible"[4,5]. Although such a perfect "PGEC" material has not been found to date, the concept inspired several strategies e.g., nanostructuring of inorganic crystals to minimize thermal conductivity towards $ZT > 1$[6]. Meanwhile, the "PGEC" concept broadened its scope with a picture that a glass-like thermal conductivity coexists with charge carriers of high mobility ($\mu$)[5,7]. In general, inorganic TE materials are brittle and either toxic or rare and are therefore unsuitable for many intriguing applications, i.e., wearable/portable devices. In stark contrast, organic TE materials are abundant, mechanically flexible and cost effective and thus provide a complementary solution to these issues.

There is no clear and strict definition of an organic "PGEC" in the current literature. By analogy to the scenario in inorganic TEs, we propose the following definition of an organic "PGEC": (i) the thermal conductivity reaches the amorphous limit of the particular material[8], and (ii) the charge carrier mobility should reach its crystalline limit. This definition has the benefit that it is phrased in terms of quantities that are easily accessible through experiments. Organic materials are likely intrinsic "phonon glasses" because of the weak van der Waals interactions between adjacent molecules[9], with good potential for maximizing temperature gradient and $ZT$ value for wearable energy harvesting[10,11]. Unfortunately, most organic TE materials are far from being "electron crystals" due to either inherently low packing order or doping-induced disorder[12–15], leading to low TE performances.

After an extensive study of p-type organic thermoelectric materials[1,3,16], the scientific community has recently turned its focus to the more challenging n-type counterparts because both efficient p- and n-type TE materials are required for practical applications. A large variety of organic semiconductors, including conjugated polymers and small molecules, have been utilized for n-type organic thermoelectrics[12,17–22]. Most n-type organic thermoelectric materials exhibit an electrical conductivity of $<2$ S cm$^{-1}$ and power factor ($S^2\sigma$) of $<10$ µW m$^{-1}$ K$^{-2}$ [23,24]. As such, the quest for "organic electron crystals" becomes key for further development of the organic TE field. Carbon nanotubes (CNTs) are able to exhibit $\sigma > 1000$ S cm$^{-1}$ upon doping with n-type dopants such as benzyl viologen and salts of crown ether complexes and thus are good candidates of "electron crystals"[25,26]. However, the typically large thermal conductivity of $>4$ W m$^{-1}$ K$^{-1}$ of CNTs excludes them from being "phonon-glasses"[25]. The fullerene $C_{60}$ exhibits lattice vibrations that are largely localized within each molecule, leading to an ultralow thermal conductivity of ~0.1 W m$^{-1}$ K$^{-1}$[27]. Furthermore, its thermal conductivity can be reduced by an alkyl side chain: Phenyl-C61-butyric acid methyl ester has a thermal conductivity that is even lower than that of $C_{60}$[28,29] due to a mismatch in the vibrational density-of-states between the buckyball and the alkyl side chain[30]. Fullerene derivatives are, therefore, expected to be excellent "phonon-glasses".

However, n-doping of fullerene derivatives is often complicated by poor host/dopant miscibility[31,32], which dramatically reduces the doping efficiency and carrier density. As a result, most incorporated dopants remain inactive, and strongly disrupt the packing of the fullerene derivative. The miscibility issue can be mitigated by increasing the host polarity via the use of a linear triethylene-glycol-type side chain[20,33]. Although this type of side-chain has proven to be beneficial for n-doping[20,34], it does not transform fullerene derivatives into 'organic electron crystals'. Incorporating a large number of mesogenic groups into fullerene derivatives is able to strengthen molecular self-assembly and thereby improve long-range packing order[35,36]. However, this technique largely dilutes the conjugated species, inevitably degrading the electronic properties.

Here, we report a molecularly n-doped fullerene derivative with meticulous design of the side chain that approaches an organic "PGEC" TE material. The key to transforming this fullerene derivative close to an "electron crystal" is the use of "arm-shaped" double-triethylene-glycol-type side chains, which enable not only efficient and thermally stable n-doping but also excellent molecular packing. This TE material exhibits an excellent electrical conductivity of $>10$ S cm$^{-1}$ but an ultralow thermal conductivity of $<0.1$ W m$^{-1}$ K$^{-1}$, leading to the best figure of merit $ZT = 0.34$ (at 120 °C) of single-host organic TE materials.

## Results

**Exploiting side-chain variations of fullerene derivatives.** Aiming for the "PGEC" concept, we varied the side chains of fullerene derivatives in order to improve molecular order. As a starting point, fullerene derivatives (PTEG-1, PPEG-1, F2A, and PTEG-2) with four different types of side chains were chosen, as shown in Fig. 1a. PTEG-1 and PPEG-1 have linear-ethylene-glycol-type side chains and differ from each other by the number of ethylene glycol units in the side chain. The former has three, whereas the latter has five ethylene glycol units in the side chain moiety. Distinctly, PTEG-2 is functionalized with "arm-shaped" double-triethylene-glycol-type side chains, which differs from the side chains of PTEG-1 in terms of both the number and geometry of the ethylene glycol units. F2A has an alkyl side chain with the same geometry as the side chain in PTEG-2.

The four types of fullerene derivatives were n-doped by solution coprocessing with 8 wt% n-DMBI. The resulting films were sequentially annealed at various temperatures for 1 h before the electrical conductivity was measured by the standard four-probe method. Figure 1b displays plots of the room-temperature electrical conductivity as a function of the annealing temperature for doped films of various fullerene derivatives. Increasing the annealing temperature from 75 °C to 120 °C, enhanced the electrical conductivity of four of the doped fullerene derivatives. This result is likely due to either local spatial arrangement of host/dopant molecules for intimate contacts, a thermally activated doping process (specific to n-DMBI[37]), or both. Further increasing the annealing temperature above 120 °C produced barely observable changes in the electrical conductivity for doped PTEG-1. By stark contrast, the electrical conductivity of the doped PTEG-2 film at room temperature started to increase when the annealing temperature was above 120 °C and reached $6.5 \pm 0.6$ S cm$^{-1}$ at an annealing temperature of 150 °C. This behavior corresponded to an enhancement in the electrical conductivity by a factor of 4 over that of the sample annealed at 120 °C. PPEG-1 has more ethylene glycol units than PTEG-1, and the electrical conductivity of the doped PPEG-1 film decreased with the annealing temperature above 120 °C. The doped F2A showed a much lower optimized conductivity of 1 S cm$^{-1}$ than the fullerene derivatives with ethylene-glycol-type side chains under the same doping conditions. However, we observed a similar conductivity enhancement after annealing above 150 °C. Given the differences in the side chains among the four fullerene derivatives, the conductivity enhancement observed in the doped

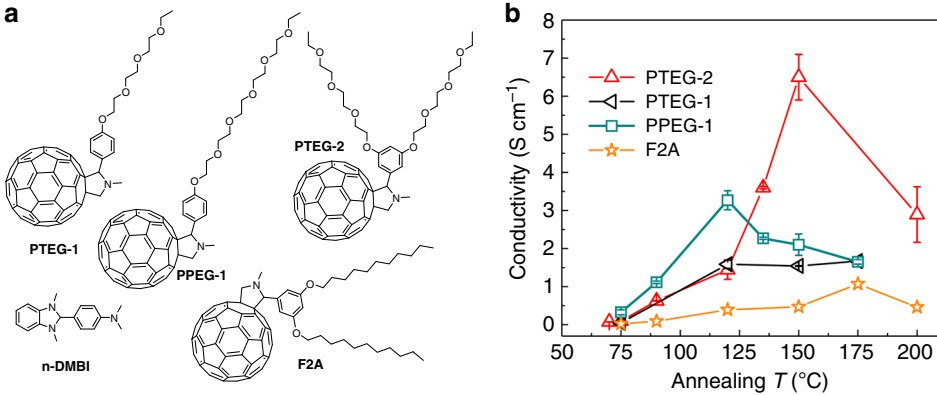

**Fig. 1 Side-chain variations and thermal annealing effect. a** Chemical structures of different fullerene derivatives (PTEG-1, PTEG-2, PPEG-1, and F2A), and dopant (n-DMBI); **b** Plots of electrical conductivity at room temperature as a function of the annealing temperature for different fullerene derivatives doped at a concentration of 8 wt% n-DMBI. Error bars indicate the standard errors of the mean values of electrical conductivity obtained by the measurement of six different samples.

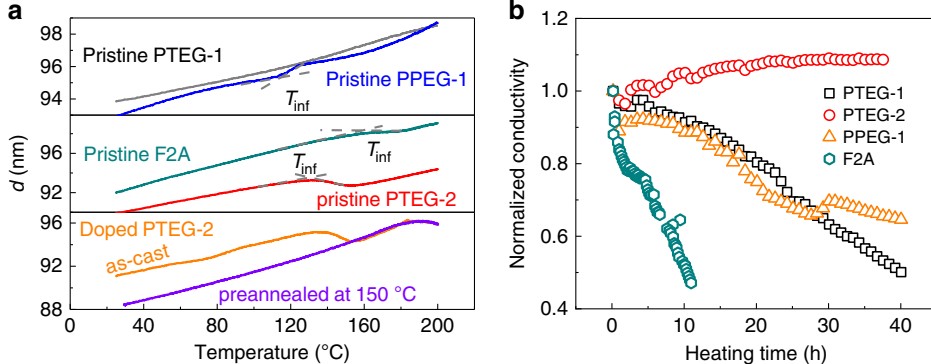

**Fig. 2 Thermal response and stability of thin-film samples. a** Plots of variable temperature ellipsometry scans for pristine PTEG-1, PTEG-2, and F2A, and doped PTEG-2 films; **b** evolution of normalized electrical conductivity for various doped fullerene derivatives maintained at a temperature of 150 °C.

PTEG-2 by annealing above 120 °C most likely stems from the unique geometry of the side chain rather than the relatively large number of ethylene glycol units of the side chain.

**In situ dynamic ellipsometry and thermal stability.** Variable temperature ellipsometry is a powerful tool for detecting the real-time phase behavior of organic films, wherein the evolution of the film thickness with temperature is followed[38]. To understand the phase behavior of the fullerene derivatives, we performed real-time dynamic ellipsometry measurements on both pristine and doped films during heating at a rate of 2.5 °C min$^{-1}$ in an N$_2$ protective atmosphere (Supplementary Note 1, Supplementary Fig. S1, and Supplementary Table S1). Figure 2a shows the film thickness ($d$) as a function of temperature ($T$) over the 25–200 °C range. The four fullerene derivatives are stable and do not exhibit any decomposition over this temperature range (Supplementary Note 2 and Supplementary Fig. S2). For a given fullerene derivative-based film that does not undergo decomposition, the film thickness scales inversely with the film density. The slope of $d$ at a certain $T$ is proportional to the linear thermal expansion coefficient. The pristine PTEG-1 film renders a nearly constant expansion coefficient upon heating. By contrast, the $d$–$T$ plot for the pristine PTEG-2 film shows an inflection point at $T_{inf} = 131$ °C, and the film even shrinks in the $T_{inf} < T < 155$ °C range. This result indicates that a phase transition occurs in the PTEG-2 film, where the film density increases at temperatures above $T_{inf}$. Such a phase transition was also observed for the doped PTEG-2

film. A phase transition was not observed in the doped PTEG-2 film that was pre-annealed at 150 °C. This result suggests that after the transition process, the phase of the PTEG-2-based film is stable after cooling down, which is technically relevant for the thermal annealing of devices. For PPEG-1, $T_{inf}$ is 115 °C, above which temperature the film expands even faster, indicating a reduced film density at higher temperatures. The phase behavior of the fullerene derivatives correlates well with the variation in the electrical conductivity with the annealing temperature. The large enhancement in the electrical conductivity of doped PTEG-2 films after annealing above 120 °C was driven by a phase transition into a more compact film, which is likely induced by the unique side chain geometry of the PTEG-2 molecule. This hypothesis was further supported by the observation of a similar phase transition at $T_{inf} = 151$ °C for F2A. The $T_{inf}$ values of the fullerene derivatives vary with the side chains in the order PPEG-1<PTEG-2<F2A. These results suggest that this specific side chain geometry, that is the 'arm-shaped' double side chains, is the main cause for the disorder-to-order transition upon thermal annealing, providing a valuable molecular design for good packing ordering.

Practical thermoelectric applications require molecularly doped organic films with adequate operational stability under thermal stress. However, most of the doped organic systems that are based on the physically blended host and dopant molecules with diverse material properties are not thermodynamically stable. Under thermal stress, the polar dopant tends to diffuse out of the host matrix and form aggregates, and this so-called de-doping process

dramatically degrades the electrical conductivity[39]. There are several reports that polar side chains could enhance the binding between host and dopant molecules and thus increase the thermal stability of doped systems[40,41]. To obtain insight into the effect of side chains on the thermal stability of doped fullerene derivatives, we probed the electrical conductivities of various doped films on a hot stage under nitrogen with a temperature of 150 °C. Figure 2b displays the corresponding evolution of the electrical conductivity over time for doped F2A, PTEG-1, PPEG-1 and PTEG-2 films. Of these materials, the ones with polar side chains are more thermally stable than the one with an alkyl side chain (F2A). The electrical conductivity of the doped PTEG-1 film drops to 50% of its original value after heating for 40 h. Increasing the number of ethylene glycol units of the side chain improves the thermal stability of the doped PPEG-1 film, and the electrical conductivity drops to 65% after heating for 40 h. In contrast, the doped PTEG-2 film does not degrade and even shows a slight increase in the electrical conductivity of 9% after heating for 38 h. This result is the first report of a completely stable molecularly doped organic film under a strong thermal stress (150 °C), which is very meaningful for practical applications.

The morphologies of pristine and doped PTEG-2 films annealed at different temperatures were analyzed by atomic force microscopy (AFM), as shown in Supplementary Fig. S3. The surface morphology of pristine PTEG-2 films shows larger domains as the annealing temperature is increased from 120 to 150 °C. Upon co-processing with n-DMBI and annealing at 120 °C, aggregates were observed on the surface of the PTEG-2 film. We propose that these aggregates are formed by unreacted dopants or some complex related to the doping process. Interestingly, upon annealing at 150 °C, these aggregates gradually disappear. As none of the species in the blend is volatile at 150 °C (Supplementary Fig. S2), we argue that more of the dopants are incorporated into the host matrix at higher temperatures. This result is opposite to the typically observed trend while annealing organic blends, wherein aggregate growth is driven on the film surface[42]. This finding indicates that well-mixed PTEG-2 and n-DMBI forms a thermodynamically stable state when annealed at 150 °C, which explains the unusually excellent thermal stability of the doped PTEG-2 film.

**Molecular packing**. The effects of thermal annealing on the microstructures of the pristine and doped PTEG-2 films were investigated by two-dimensional grazing incidence wide-angle X-ray scattering (2D-GIWAXS) (see details in Supplementary method 1). Figure 3a–d shows the GIWAXS patterns and linecuts of pristine PTEG-2 films annealed at 120 and 150 °C. For both samples, three strong reflections focused along the near out-of-plane $q_z$ direction were visible in the low angle region. These signals corresponded to the (00l) family of reflections, suggesting that PTEG-2 mainly adopts a layered structure along the substrate normal direction. The spacing extracted from the (001) peak position for the pristine PTEG-2 thin film is 2.7 nm and is not affected by the annealing process. Annealing at a higher temperature (i.e., 150 °C) leads to a significant increase in the (00l) intensities along $q_z$ and a decrease along the $q_y$ direction as a result of a reduction in the angular spreading of the (00l) reflections. This result suggests an increased orientation ordering of the crystals due to the annealing process at $T = 150$ °C. Along the in-plane $q_y$ direction, we observed peaks at $q_y = 0.22$ and 1.25 Å$^{-1}$ for the pristine PTEG-2 film annealed at 120 °C, which correspond to a (001) signal associated with an interplanar distance of 2.83 nm, and a (020) peak, which is associated with a spacing of 0.50 nm, respectively. The dimension of the c-axis measured by GIWAXS suggests that the ethylene glycol type side chains are oriented vertically with respect to the layer plane that contains the fullerene molecules. Interestingly, after annealing at 150 °C, the (001) peak in $q_y$ direction is considerably suppressed, and the (020) peak is considerably enhanced, because of the increased crystal orientation. These results indicate that annealing at a higher temperature predominately impacts the molecular order along the in-plane direction, which is highly relevant for charge transport. Moreover, upon annealing at 150 °C, clear off-specular and out-off-plane peaks appear, suggesting that the molecules are packed into a more efficient crystalline structure. A similar trend is observed for the doped PTEG-2 samples (Supplementary Fig. S4). The increased crystallinity driven by annealing at a higher temperature agrees well with the phase transition process detected by the spectroscopic ellipsometry, resulting in a plausible explanation of the origin of the enhanced electrical conductivity.

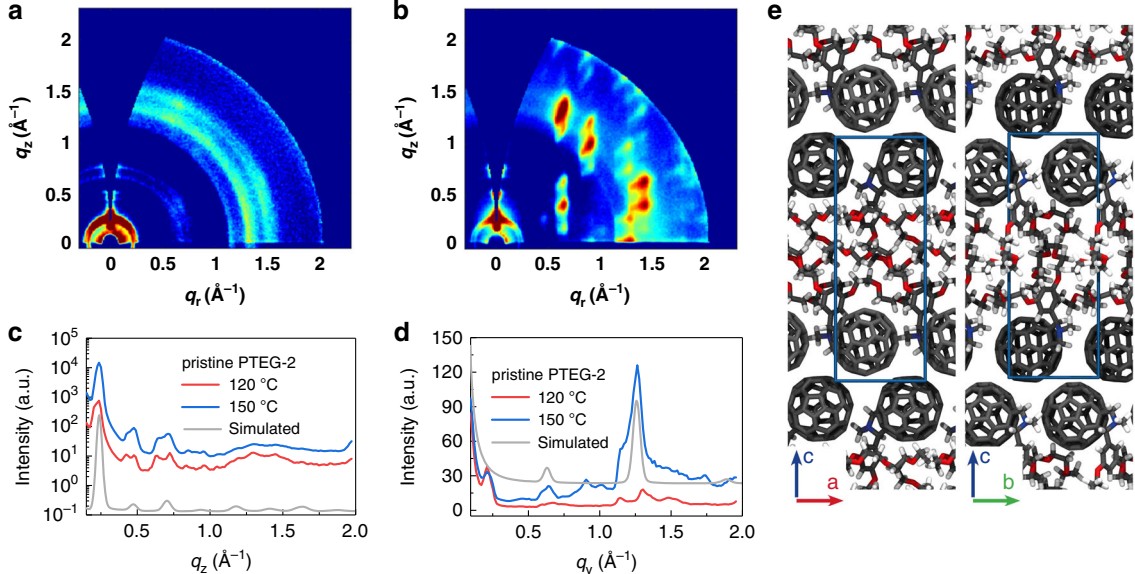

**Fig. 3 Molecular packing of PTEG-2 films. a, b** 2D-GIWAXS patterns of pristine PTEG-2 films annealed at (**a**) 120 °C and (**b**) 150 °C and (**c, d**) the corresponding linecuts together with the simulated scattering linecuts (the simulated linecuts are plotted on a linear scale in both cases); **e** representative snapshot of PTEG-2 molecular packing, as atomically resolved by molecular dynamics simulations; the unit cell is highlighted in blue.

We resolved the molecular packing in the unit cell of PTEG-2 at atomistic level using molecular dynamics (MD) simulations. Based on the layered structure inferred from the 2D-GIWAXS data, a bias is imposed on the MD simulations for the fullerene moieties to be in contact. Other than that, the three unit cell parameters, $a$, $b$, and $c$, are able to fully relax during the MD simulations, and we thus obtain distributions of the final cell parameters as the outcome of 240 simulations (see "Methods", Supplementary Note 3 and Supplementary Fig. S5 for details). The resulting average $c$-axis of $2.72 \pm 0.01$ nm is in very good agreement with the $c$-axis obtained from the 2D-GIWAXS measurements ($2.70 \pm 0.02$ nm). The $a$ and $b$ axes are $1.01 \pm 0.01$ nm, accommodating $C_{60}$ and the slightly tilted pyrrolidine moiety. The simulated scattering linecuts (Fig. 3c, d, gray lines) further confirm the resolved unit cell: the (001) family of reflections is visible in the low angle region along $q_z$, while the (002) peak is clearly visible along $q_y$. A few characteristics of the experimental spectra are missing in the simulated scattering linecuts. This is likely due to the fact that the simulated crystals are perfect, whereas the experimental morphology includes some level of disordered and misaligned crystal domains. A sample representation of the unit cell, taken from among the lowest energy configurations that are computed by periodic density functional theory calculations, is shown in Fig. 3e. A common feature of the multiple configurations obtained from the MD simulations is a staggered arrangement for the $C_{60}$ bilayers interposed by the ethylene glycol phase. However, the MD simulations do not converge all into one specific configuration of the ethylene glycol chains, but rather give an ensemble of similar ones. The convergence into a single ethylene glycol configuration may have been prevented either by the limited simulation time (also compared to the experimental annealing times) or by the fact that the ethylene glycol chains are quite flexible due to the low energy barriers between their different configurations[43].

**Tuning doping level for power factor optimization.** We optimized the power factor of the doped PTEG-2 films by systematically tuning the doping level via the n-DMBI loading. Various doped PTEG-2 films were annealed at 120 °C and 150 °C, respectively. Figure 4 displays the in-plane thermoelectric parameters as a function of doping concentration for the doped PTEG-2 films (see details in Supplementary Fig. S6). As shown in Fig. 4a, doping PTEG-2 produces the highest electrical conductivity of $8.3 \pm 0.5$ S cm$^{-1}$ at a doping concentration of 5 wt%

upon annealing at 150 °C. By tuning the doping concentration from 0.5 to 10 wt%, the absolute value of the Seebeck coefficient of the doped PTEG-2 samples annealed at 150 °C decreases from $-(731 \pm 35)$ to $-(163 \pm 10)$ µV K$^{-1}$ (Fig. 4b). Interestingly, although the electrical conductivity of doped PTEG-2 annealed at 150 °C is approximately four times that of the samples annealed at 120 °C, the Seebeck coefficient appears to be only slightly influenced by the annealing temperature. The highest power factor of $41 \pm 5$ µW m$^{-1}$ K$^{-2}$ was achieved at room temperature for the 5 wt%-doped PTEG-2 film annealed at 150 °C.

It is interesting to find that the doped PTEG-2 film is as conductive as some of the alkali metal doped $C_{60}$ films reported in the literature (4 S cm$^{-1}$ for Cs-doped $C_{60}$, 10 S cm$^{-1}$ for Li-doped $C_{60}$, and 20 S cm$^{-1}$ for Na-doped $C_{60}$)[44]. The K-doped $C_{60}$ film, however, has an even higher conductivity of 500 S cm$^{-1}$ but has a much smaller Seebeck coefficient ($S = -11$ µV K$^{-1}$)[45] than doped PTEG-2. We directly measured the free carrier densities ($n$) of the doped PTEG-2 films by using admittance spectroscopy on ion-gel-based metal insulator semiconductor devices (see details in Supplementary method 2 and Supplementary Fig. S7a, b). The 5 wt%-doped PTEG-2 films upon thermal annealing at 120 and 150 °C exhibit free carrier densities of $(3.5 \pm 0.2) \times 10^{19}$ cm$^{-3}$ and $(4.5 \pm 0.3) \times 10^{19}$ cm$^{-3}$, respectively, which are within the optimal regime ($10^{19}-10^{20}$ cm$^{-3}$). The formula $\sigma = n \times \mu \times q$ is used to calculate a bulk mobility of the doped film of ~0.37 cm$^2$ V$^{-1}$ s$^{-1}$ upon annealing at 120 °C and ~1.2 cm$^2$ V$^{-1}$ s$^{-1}$ upon annealing at 150 °C. Ogata et al. found a (time-of-flight) mobility of $0.5 \pm 0.2$ cm$^2$ V$^{-1}$ s$^{-1}$ for single-crystal $C_{60}$ and Anthopoulos et al. reported a highest field-effect transistor mobility of 6 cm$^2$ V$^{-1}$ s$^{-1}$ for hot-wall epitaxy grown crystalline $C_{60}$ film[46,47]. It stands to reason that the charge mobility of $C_{60}$ is an upper limit to that of fullerene derivatives as the side chains dilute the conjugated moieties. The high bulk mobility (>1 cm$^2$ V$^{-1}$ s$^{-1}$) is of the same order of magnitude as that of single crystal $C_{60}$, indicating that the doped PTEG-2 comes close to the mobility requirement of an organic "electron crystal".

The doping efficiency is defined as the ratio of free carrier density to the number of dopant molecules. We calculated the doping efficiency in doped PTEG-2 film and obtained a value of ~47% after annealing at 120 °C and ~60% after annealing at 150 °C with an estimated total site density $3.7 \times 10^{20}$ cm$^{-3}$ from the lattice structure. The electron paramagnetic resonance spectra (Supplementary Fig. S7c) indicate that 40% more polarons are generated upon annealing at 150 °C instead of 120 °C, resulting in the improved doping efficiency. This is consistent with the

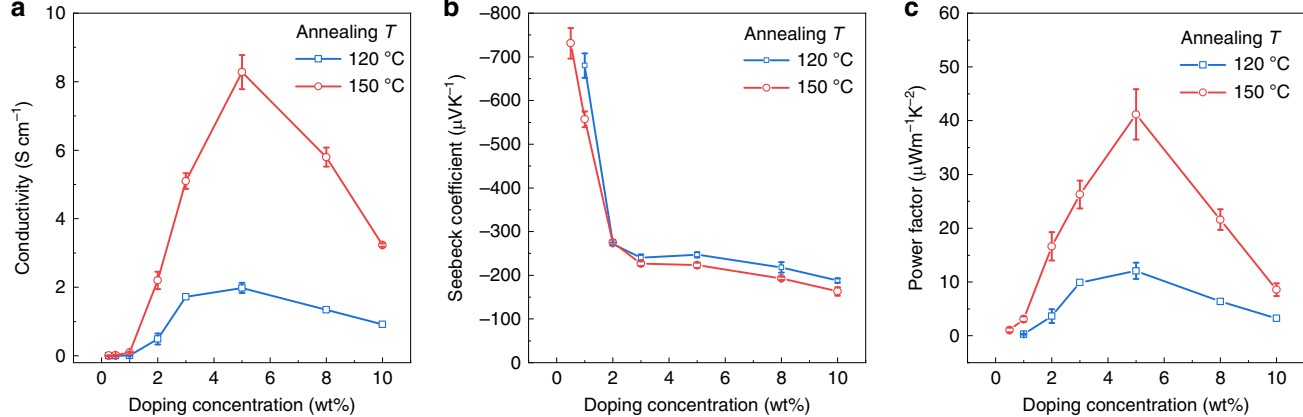

**Fig. 4 Optimization of thermoelectric parameters by controlling the doping. a** electrical conductivity, **b** Seebeck coefficient, and **c** power factor as a function of doping concentration for doped PTEG-2 film at room temperature. Error bars indicate the standard errors of the mean values of electrical conductivity, Seebeck coefficient and power factor obtained by the measurement of six different samples.

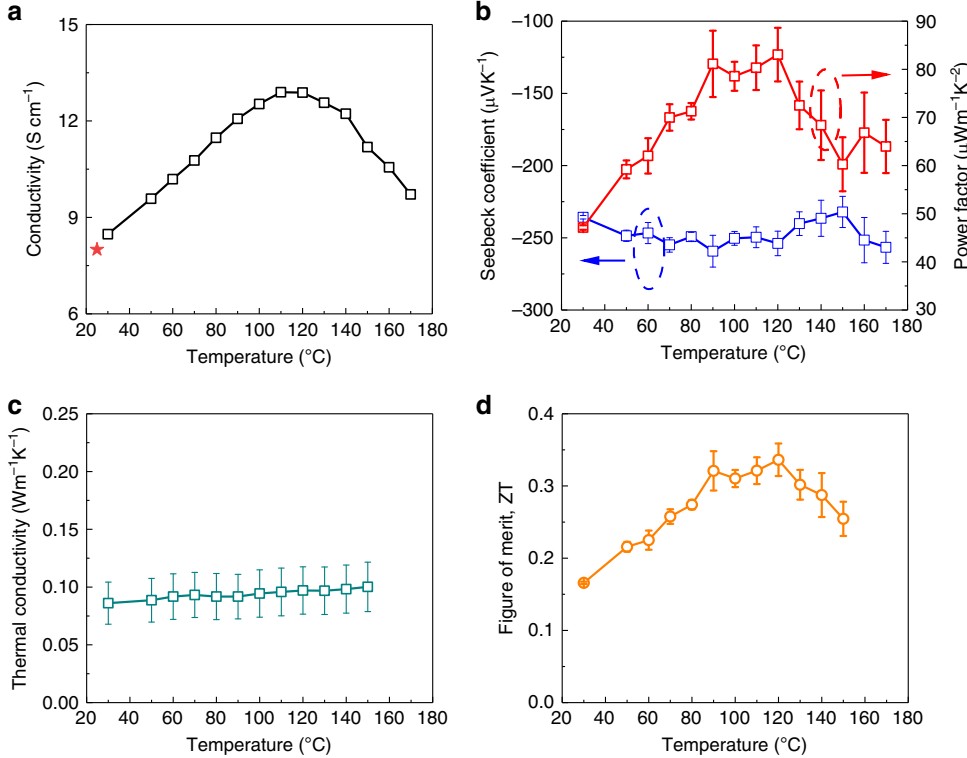

**Fig. 5 Temperature dependent thermoelectric parameters. a** electrical conductivity (the red star represents the conductivity after cooling down to 25 °C), **b** Seebeck coefficient (blue) and power factor (red), **c** in-plane thermal conductivity and **d** figure of merit, *ZT* at various operating temperatures for the doped PTEG-2 film at a 5 wt% doping concentration. Error bars (**b**, **c**) represent the standard errors of Seebeck coefficient and thermal conductivity obtained by best-fits; error bars (**d**) represent the corresponding calculated deviations of *ZT*.

improved mixing between the host and the dopant molecules as revealed by the morphology study.

We measured the in-plane thermal conductivity ($\kappa_\parallel$) for the pristine and doped PTEG-2 films after annealing at 150 °C by the standard 3-omega (3ω) method (see details in Supplementary Note 4 and Supplementary Fig. S8). The pristine PTEG-2 film displays a very low $\kappa_\parallel$ of 0.064 W m$^{-1}$ K$^{-1}$ at room temperature, which is among the lowest reported thermal conductivities for a fully dense solid. This low thermal conductivity is attributed to the localized lattice vibration within each molecule[27]. After doping with 5 wt% n-DMBI, the $\kappa_\parallel$ of PTEG-2-based film slightly increases to 0.086 W m$^{-1}$ K$^{-1}$. This value is still very low with respect to recently reported values (>0.2 W m$^{-1}$ K$^{-1}$) for doped conjugated polymers or small molecules[22,48]. By following an empirical Lorenz number ($L$)-Seebeck coefficient relation for non-degenerate semiconductors[49], we estimated $L = 1.65 \times 10^{-8}$ W Ω K$^{-2}$ for the doped PTEG-2 film. The electronic contribution ($K_E$) to the thermal conductivity was thus estimated to be minor (<0.01 W m$^{-1}$ K$^{-1}$), indicating that the lattice thermal conductivity is dominant in the doped PTEG-2 films. We explored anisotropy effects by measuring the out-of-plane thermal conductivity ($\kappa_\perp$ via time-domain thermoreflectance (TDTR))[50,51] (see details in Supplementary Note 5 and Supplementary Fig. S9). We extracted a value of $\kappa_\perp = 0.069$ W m$^{-1}$ K$^{-1}$ for the doped PTEG-2 film, indicating low anisotropy in the thermal transport. This result agrees very well with previously reported out-of-plane thermal conductivities (0.05–0.06 W m$^{-1}$ K$^{-1}$) for various fullerene derivatives (measured by TDTR as well)[29]. These results indicate that the thermal conductivity of the doped PTEG-2 is even lower than the amorphous limit of this type of material[8,29]. This qualifies the doped PTEG-2 as an excellent organic "phonon glass". Such an organic "phonon glass" is very attractive for wearable applications as it sustains a large temperature gradient when in contact with the

human body[11], making doped PTEG-2 very promising for wearable energy harvesting. We used the measured power factor and $\kappa_\parallel$ to determine a figure of merit of $ZT = 0.15$ for the doped PTEG-2 film at room temperature.

**Temperature-dependent thermoelectric parameters.** One of the promises of organic thermoelectrics is the efficient energy conversion of low-temperature (<250 °C) waste heat. Thus, it is important to obtain insight into the thermoelectric performance at elevated temperatures. Figure 5 displays the variable temperature thermoelectric parameters of the doped PTEG-2 film (see details in Supplementary Note 6 and Supplementary Fig. S10). Varying the temperature from 30 to 110 °C increased the electrical conductivity from 8.5 to 12.9 S cm$^{-1}$ while further increasing the temperature above 120 °C resulted in a decrease in the electrical conductivity. After cooling down to 25 °C, the electrical conductivity was back to 8.0 S cm$^{-1}$ (Fig. 5a, red star), very close to its original value. This result agrees with the excellent thermal stability of the doped PTEG-2 and thus excludes de-doping as the cause of the conductivity drop above 120 °C. The small temperature dependence of the electrical conductivity with an activation energy of 51 meV (~2 $k_B T$) (Supplementary Fig. S11) below 110 °C suggests a nearly disorder-free charge transport in the doped PTEG-2 film. As for the decreased conductivity above 120 °C, we speculate that, at higher temperatures, intensified molecular vibrations hamper charge transport, analogously to phonon scattering in an inorganic crystal. The features of the temperature dependent electrical conductivity together with the high charge mobility of >1 cm$^2$ V$^{-1}$ s$^{-1}$ indicate that doped PTEG-2 is close to an organic "PGEC" material. By contrast, the Seebeck coefficient appears to vary only slightly with temperature. A very high power factor of over 80 µW m$^{-1}$ K$^{-2}$ is

realized at the temperature range of 90–120 °C. Like typical glassy materials, the in-plane thermal conductivity of doped PTEG-2 exhibits a very small temperature dependence and slightly increases to 0.097 W m$^{-1}$ K$^{-1}$ at 120 °C. The optimized $ZT$ is 0.34 at 120 °C, which is the highest reported result among any single-host organic thermoelectrics.

## Discussion

In conclusion, we have applied the "phonon-glass electron-crystal" concept to n-type organic thermoelectrics and varied the side chains of "phonon-glassy" fullerene derivatives in order to realize an electron crystalline film. We find that "arm-shaped" double-triethylene-glycol-type side chains with a unique geometry are able to not only offer efficient and thermally stable n-doping of the fullerene derivative but also induce a disorder-to-order transition upon thermal annealing over a certain temperature. Therefore, the n-doped fullerene derivative is transformed close to an "organic electron crystalline" film (with $\sigma > 10$ S cm$^{-1}$ and $\kappa < 0.1$ W m$^{-1}$ K$^{-1}$), which exhibits the best $ZT = 0.34$ (at 120 °C) among reported single-host organic TE materials. This work is a proof-of-concept study of how to apply the PGEC concept to organic TEs and sheds light on the molecular design toward an "organic electron crystal" for high-$ZT$ thermoelectrics. Furthermore, the doped PTEG-2 with an ultralow $\kappa$ and excellent $\sigma$ could be potentially used to form composites with other promising TE materials (such as CNTs and inorganic crystals) for tunable TE properties.

## Methods

**Computational methods**. Using the layered structure inferred by the GIWAXS data as a basis, MD simulations were carried out to obtain atomistic configurations for the two molecules in the unit cell. Periodic boundary conditions were applied in three directions. The MD simulations were carried out in several steps to maximize sampling, with a total simulation time of 13 ns per run. The three unit cell parameters, $a$, $b$, and $c$ were able to fully relax during the MD simulations. The simulations were repeated 240 times to yield the final cell parameter distributions (see Supplementary Fig. S5). Note that the final cell parameters were obtained from simulation boxes (of size of about $5 \times 5 \times 8$ nm$^3$) containing multiple unit cells. The MD cell parameters and the corresponding errors were extracted as the mean and standard deviation of the cell parameter distributions, respectively. The PTEG-2 force field necessary for the MD simulations was realized by adding an extra identical TEG chain to the nonpolarizable PTEG-1 force field recently developed by Sami et al.[43]. The MD simulations were performed using the GROMACS 2016.x software package[52]. The detailed settings are provided in Supplementary Code File. Periodic density functional theory calculations using the CRYSTAL software[53] were performed on the 125 out of the 240 final MD conformations that were within 0.05 nm of the MD average $c$-axis of 2.72 nm. The PBE functional, a 6–31 G** basis set and 36 k-points were used. A sample CRYSTAL input file is included in the Supplementary information. The simulated scattering linecuts have been computed by Fourier transform of the atomic coordinates provided by the MD simulations along the $z$ ($c$ axis) and $y$ ($b$ axis) dimensions. The linecuts represent an average of the 240 MD unit cells. The detailed procedure for the calculation of the linecuts is given in Supplementary Note 3.

**Preparation of n-doped fullerene derivative films**. PTEG-1, PPEG-1, F2A, and PTEG-2 were synthesized according to a previously reported procedure[54]. The n-DMBI dopant was purchased from Sigma Aldrich. The doped fullerene derivatives films were prepared by spin-coating PTEG-1, PPEG-1, F2A, and PTEG-2 solutions (10 mg ml$^{-1}$ in chloroform) mixed with different amounts of n-DMBI dopant solution (20 mg ml$^{-1}$ in chloroform) in a glove box under a nitrogen atmosphere. The prepared blended films were thermally annealed at various temperatures. The thicknesses of the various organic films were measured by AFM and ellipsometry.

**Thermoelectric parameters at room temperature**. Clean borosilicate glass substrates were treated with UV-ozone for 20 min. Four parallel line-shaped Au electrodes (Au line-width: 0.5 mm; channel width: 4.5 mm; and channel length: 1 mm) for the 4-point probe electrical conductivity measurement and two parallel line-shaped Au electrodes (Au line-width: 1.5 mm; channel width: 6 mm; and channel length: 6 mm) for the Seebeck coefficient test were deposited on the clean glass substrates. Various doped fullerene derivative films were prepared on the electrode-coated substrates. The area of the doped organic films outside the region defined by the electrodes was scratched to prevent geometric effects. The four-point-probe measurements were performed using a Keithley 4200 SCS parameter

analyzer in an N$_2$-controlled environment. The $I$–$V$ plots are displayed in Supplementary Fig. S6. The electrical conductivity was calculated using $\sigma = (I/V)L/(wd)$, where $L = 1$ mm, $w = 4.5$ mm and $d$ denote the channel length, width and thickness of the doped organic film, respectively. The reported values in the main text were obtained by averaging the results for six devices. The Seebeck coefficients of the various doped fullerene derivative films were measured using a setup that was developed in-house, as reported previously[20]. Two Peltier devices were placed in parallel separated by a 6-mm gap on a heat sink. The temperatures of these devices were controlled by two independent proportional-integral-derivative controllers. Two rectangular Au electrodes (width: 1.5 mm and length: 6 mm) were deposited on the glass substrate separated by a distance of 6 mm. Two T-type thermocouples (127 m from Omega) were used as probes to simultaneously measure the temperature and thermal voltage. A silver paste (ELECTRODAG 1415) was used to connect the thermocouple probes with the Au electrodes. A Keithley 2000 mounted with a scanner card was used for signal recording with a delay time of 100 ms. The system was controlled by Labview software. Thermal voltage ($V$) shifts were eliminated using a quasi-static approach by slowly changing the temperature: the temperature difference ($T$) was used to extract a $T$–$V$ plot, and a linear fitting was performed to derive the Seebeck coefficient. The thermal conductivity was measured using a Linseis thin film analyzer setup employing the 3-omega (3ω) method[55].

**Thin film characterization**. AFM topographical images were recorded in ScanAsyst mode using a Bruker MultiMode 8 microscope using ScanAsyst-Air probes. The phase behavior of the organic films was investigated by spin-casting pristine and various doped fullerene derivative films on clean silicon substrates with a thin layer of native oxide. The thin-film samples were placed in an air-protected sample holder under a continuous N$_2$ flow, the sample holder was mounted on a temperature-controlled stage, and optical measurements were carried out using variable angle ellipsometry (J. A. Wollam Co., Inc). The ellipsometry data was collected while programmatically heating the sample to probe the phase behavior of the organic film (details can be found in Supplementary Note 1 and Supplementary Fig. S1).

## Data availability

The data that support the findings of this study are available from the corresponding authors upon request.

## Code availability

The code scripts needed to reproduce the MD simulations that support the findings of this study are available as Supplementary code file. The code used to obtain the simulated scattering linecuts is available from the corresponding authors upon request.

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

## Acknowledgements

This study was supported by a grant from STW/NWO (VIDI 13476). This study is carried out under the auspices of the research program of the Foundation of Fundamental Research on Matter (FOM), which is part of the Netherlands Organization for Scientific Research (NWO). This is a publication by the FOM Focus Group "Next, Generation Organic Photovoltaics", participating in the Dutch Institute for Fundamental Energy Research (DIFFER). J.D. acknowledges financial support from the China Scholarship Council. L.Q. thanks National Natural Science Foundation of China (Grant No. 51962036) for financial support. D.B. acknowledges financial support from a KAUST Competitive Research Grant (3737 GRG7). Computational resources for this work were partly provided by the Dutch National Supercomputing Facilities through NWO. R.A. and S.S. thank Anna S. Bondarenko and Jordi Antoja-Lleonart for insightful discussions on the simulated scattering spectra.

## Author contributions

J.L. and L.J.A.K. conceived this study. J.L. and B.Z. characterized the thermoelectric properties and phase behavior of the materials. J.L. measured the carrier density and thermal stability of the materials. J.D. and G.P. obtained and analyzed the GIWAXS data. R.A. and S.S. performed and analyzed the molecular dynamics, quantum mechanical, and scattering simulations under the supervision of R.W.A.H and S.J.M. M.I.N. measured the in-plane thermal conductivity under the supervision of D.B. M.I.N measured the EPR of the fullerene derivative based films under the supervision of T.D.A. X.Q. performed AFM under the supervision of R.C.C. A.B. performed the measurement and analysis of the thermal conductivity in the vertical direction under the supervision of M.C. L.Q., S.R., and N.K. prepared fullerene derivatives under the supervision of J.C.H. J.L. and L.J.A.K. supervised the project. J.L. and L.J.A.K. wrote the draft manuscript and all of the authors reviewed the manuscript.

## Competing interests

The authors declare no competing interests.
