## [Peer review file · Nature Communications]

Reviewers' Comments:

Reviewer #1:

Remarks to the Author:

The authors carry out an excellent study of the thermoelectric properties of a series of fullerene-derivatives. The striking result is a record high ZT for n-type organic materials (and perhaps higher than the p-type record).

The work is very thorough and I do not have any major comments on the detailed technical work.

A minor point is that I do not really think this is an electron-crystal/phonon glass. There is no evidence of extended state conduction, i.e. an electron crystal. Likely the conduction is by hopping which, to me, is not really an electron crystal. The paper cited on disorder is for a polymer that arguably has higher disorder, but a much higher electrical conductivity which somewhat contradicts the statement. This is not meant to detract from the major results in the paper.

Can the authors compare these results to other doped fullerenes, e.g. potassium-doped materials, in literature to determine the role of the functional group? It would be nice to see the Lorenz factor here; the electrical conductivity is likely too low for the thermal conductivity of electrons to have much of an impact.

Reviewer #2:

Remarks to the Author:

The manuscript entails an experimental study of the thermoelectric performance of a molecularly doped fullerene derivative. A branched ethylene glycol side chain is found to result in a, for fullerenes, record electrical conductivity while preserving a low thermal conductivity. Overall, a for organic materials very high thermoelectric figure of merit is obtained; the value is among the highest values reported to date. The thermoelectric data are certainly very good, and should be published in a journal such as Nature Communications. However, the analysis of the data both in terms of (heat and electrical) transport physics and physical chemistry needs improvement. Therefore, I cannot recommend publication of the manuscript in its current form. I am sorry for not being more positive, but hope that my comments below will help the authors to further sharpen their manuscript.

1) My major concern is the proposition that the presented material is a phonon-glass electron-crystal. While this argument is feasible, the manuscript lacks proof that could substantiate such a claim. As the authors state, this insight would be a considerable advance for the field of organic semiconductors. At a minimum, the authors should present temperature dependent thermal and electrical conductivity data. That should allow to identify phonon-glass electron-crystal type behavior.

2) the authors mention the doping efficiency repeatedly. I recommend that the authors define this term more carefully. Do the authors refer to a charge transfer event? Not each generated polaron will lead to a free charge. Ionization and dissociation of charges should be discussed.

3) in the introduction the authors write that organic materials are not toxic. That is possibly true for the semiconductors. However, molecules like DMBI are highly reactive. The SDS provided by Sigma states: acute toxicity. I suggest that the authors reconsider their claim.

4) line 121. The author speculate that the fullerene material changes its "arrangement" upon annealing. The authors should provide evidence that no solvent is trapped in the films. Even chloroform may be trapped, despite a low boiling point, which could explain the observed decrease in film thickness.

5) The film thickness during annealing is extracted from ellipsometry measurements. Detailed information about the model should be provided that was used to convert the ellipsometric angles into film thickness. How were the raw data fitted and how was the accuracy of the fit determined? At minimum, a least square analysis of the fit should be provided. And what range of wavelengths were fitted?

6) The authors conclude from their measurements that the fullerene materials undergo a phase transition, and propose that side chain melting occurs. Line 171: " T_{∞} can be physically interpreted as the melting point of the side chains". I am not quite sure what a "side chain melting" is supposed to mean. In any case, a melting event would lead to a sudden decrease in density, and therefore an increase in thickness. The authors observe a decrease in thickness though. In my opinion, the suggested thermal behavior is pure speculation and must be substantiated with other types of measurements.

7) related to 6), the comment on line 174: "melting...provide[s] energy for rearrangement" makes no sense. How does melting provide energy? Do the authors mean the heat that is released by a melting event? Please measure that heat.

8) line 171. The authors insinuate that the polarity of side chains improves the thermal stability. Only one of the polar side chains (those of PTEG-2) show good thermal stability. Surely, it is not the polarity but some other parameter that determines the stability. That aside, I agree with the authors that their observation of a thermally stable doped system is significant.

9) line 200. "aggregates gradually disappear. How is the presence and disappearance of aggregates included in the ellipsometric model that was used to determine the film thickness. Did the authors assume a multilayer stack? If the model does not include at least a bilayer (the top layer represents the aggregates) that disappears upon annealing, then the observed changes in thickness could easily be an artefact. I must thus question the ellipsometry analysis.

10) Figure 3 and paragraph starting at line 235. The MD simulations are very illustrative. I would like to see a simulation of the diffraction pattern in Fig 3b, which substantiates the proposed unit cell.

11) line 249. Now the ethylene glycol side chains are described as having different configurations. How is this consistent with the picture provided earlier where these side chains order at low temperatures, and upon heating?

12) line 272. A doping efficiency of 60% is calculated. Are these polarons formed per dopant molecule or free charges? And how does this value compare to the proposed disappearance of aggregates (line 200), which leads to a higher conductivity. Does the doping efficiency increase upon annealing? The authors quote carrier densities at 120 and 150 deg but only one doping efficiency. Is this because the doping efficiency does not actually increase, which would be inconsistent with the earlier argument that more dopant is taken up by the fullerene material?

13) line 298. Now the side chain melting occurs at 120 deg. On line 169 it is 98 deg.

Minor comments:

1) line 70. Explain "proper n-doping"

2) line 259 and 267. Provide error bars for the Seebeck coefficient and carrier densities.

3) The doped fullerene material is not stable under ambient conditions. All electrical characterization was done under nitrogen. But how about the thermal conductivity measurement.

How did the authors ensure that the sample had not degraded? The Linseis system requires that the sample is handled in air. Was the instrument placed in a glove box?

Reviewer #3:

Remarks to the Author:

Liu et al. claimed that they developed OTE materials with ZT value over 0.3 following PGEC concept. In fact, this concept is widely used in inorganic materials, but its applications in OTE materials remain unclear. Although the basic concept seems interesting, I have the follow concerns about the results.

(1) For typical PGEC concept in inorganic materials, the designed atoms are introduced into crystal cell to ensure high electrical conductivity and suppress the lattice thermal conductivity. Although the authors claimed their designed materials is similar the concept, I failed to find a clear definition of the concept in organic materials. Notably, several doped organic semiconductors have been confirmed to possess ordered molecular packing and the dopant is located at side-chain regime. Once again it raises a question, how could define PGEC materials in conjugated systems? Whether these previous reported materials also follow the concept?

(2) The ultralow thermal conductivity and high ZT are the key evidences for the so-called PGEC concept. I have several concerns about measurement of the κ and ZT values:

i) As a well-known fact, it is challenge to measure sample with low in-plane thermal conductivity ($\kappa \leq 0.1$ W/m K). Even with the technique provided in the supporting information, I'm not sure about the accuracy of results. A preferred thickness of the sample should be > 400 nm for a sample with κ of 0.4 W/m K (Journal of Electronic Materials 2018, 47 (6) , 3203-3209. DOI: 10.1007/s11664-017-5989-4). I believe that the situation is very challenge for PTEG-2 film since much thicker film is required. The author should provide the exact thickness of the sample used in TFA measurement and evaluate the deviation.

ii) I note the measurement of σ and S were performed in a N₂-controlled environment while $\kappa_{||}$ was obtained in vacuum by TFA. What about the air stability of the sample during the sample transfer? Such kind of n-type materials usually possess poor air stability, which will lead to rapid dedoping and lower $\kappa_{||}$ because of reduced contribution from electron transport. In fact, TFA can characterize all the parameters and even temperature dependent performance in one chip. Why the author prefers to utilize different techniques to characterize the samples?

iii) It is difficult to understand that the σ shows a large variation versus temperature ($>30\%$) while the $\kappa_{||}$ almost keeps nearly unchanged in Figure 5. The electronic contribution of the κ is related to the σ and follows Wiedemann-Franz law. In other words, one should observe the temperature dependent κ even for PGEC materials. The authors should identify the electronic contribution and lattice contribution to the κ to confirm and understand the phenomenon.

3) The authors found that the electrical conductivity first rises then falls along with the increase temperature. They attribute the decreased σ to the enhanced disorder of the side chain. Providing the energetic disorder before and after phase transition will make it clearer. In fact, dedoping can also contribute to the decrease of the conductivity. What about the thermal stability of doped films?

4) The doped PTEG-2 film shows high crystallinity. Since the author use 'electron crystal' to describe the materials, I suggest the authors to evaluate of the carrier concentration and mobility by TFA to support the definition.

Based on these concerns, I am afraid that I cannot recommend its publication in the present form.

We thank the reviewers for their constructive comments and positive appraisal of our manuscript. We have performed a number of additional experiments and have added text and plots to address the issues raised by the reviewers.

Reviewer #1:

The authors carry out an excellent study of the thermoelectric properties of a series of fullerene derivatives. The striking result is a record high ZT for n-type organic materials (and perhaps higher than the p-type record). The work is very thorough and I do not have any major comments on the detailed technical work.

1). A minor point is that I do not really think this is an electron-crystal/phonon glass. There is no evidence of extended state conduction, i.e. an electron crystal. Likely the conduction is by hopping which, to me, is not really an electron crystal. The paper cited on disorder is for a polymer that arguably has higher disorder, but a much higher electrical conductivity which somewhat contradicts the statement. This is not meant to detract from the major results in the paper.

Response:

We thank the review for the nice comments. After carefully considering the reviewer's point, we drop the claim that we fully realized an organic 'PGEC' in the present study, and instead claim that we are close to it.

Unfortunately, there is no clear, strict definition of an organic PGEC in the current literature. Even for an inorganic PGEC it is difficult to find a proper definition. Therefore, in the revised version of our manuscript, we propose the following definition of an organic PGEC:

- *the thermal conductivity reaches the amorphous limit (<https://journals.aps.org/prb/abstract/10.1103/PhysRevB.46.6131>) of this material;*
- *the charge carrier mobility should reach the crystalline limit of this particular material.*

In this manuscript we show that the thermal conductivity does indeed reach (it's actually even lower than) the amorphous limit. For the related fullerene derivative PCBM there are several reports in the literature that show that the pristine material (i.e. undoped) does indeed satisfy this part of the definition. Here we show that 1) PTEG-2 has a similarly low thermal conductivity, and 2) it is not affected by doping. In sum, doped PTEG-2 is a phonon glass.

Regarding the charge carrier mobility: Frankevich et al. (Frankevich, E.; Maruyama, Y.; Ogata, H. Chem. Phys. Lett. 1993, 214, 39.) have found a (time-of-flight) mobility of 0.5 ± 0.2 cm²/Vs for single-crystal C60 and Bao et al. reported a high field-effect transistor mobility of 5.2 ± 2.1 cm²/Vs for a needle-like single crystal (J. Am. Chem. Soc. 2012, 134, 2760). It stands to reason that the mobility of C60 is an upper limit to that of fullerene derivatives as the side-chains will dilute the conjugated part of the system. In our manuscript, we find a bulk mobility of 1.2 cm²/Vs which is on the same order of magnitude as that of single crystal C60. In other words, PTEG-2 comes close to (if not actually fulfills) the mobility requirement.

The referee is right, however, in saying that doped PTEG-2 probably does not show band transport. We do note, however, that the conductivity (see Fig 5a) shows little temperature dependence. Above 120 °C the conductivity even shows a negative temperature dependence. Figure 5b (Seebeck coefficient) shows that the Seebeck coefficient does not change with temperature, suggesting that the number of charge carriers remains constant. Taken together, this suggests that the conductivity might even be metallic in the sense that it shows negative temperature dependence. However, we do not wish to claim that this is the case as that would require a more detailed study.

All in all, we have revised our claim by stating that our material system approaches the PGEC concept.

2) Can the authors compare these results to other doped fullerenes, e.g. potassium-doped materials, in literature to determine the role of the functional group? It would be nice to see the Lorenz factor here; the electrical conductivity is likely too low for the thermal conductivity of electrons to have much of an impact.

Response:

We compared the best doped PTEG-2 film with reported alkali metal doped C60 films in terms of electrical conductivity. It is interesting to find that the electrical conductivity ($>8 \text{ S cm}^{-1}$) for the 5 wt%-doped pteg-2 film is within the conductivity range of various alkali metal doped C60 films (4 S cm^{-1} for Cs-doped C60, 10 S cm^{-1} for Li-doped C60, 20 S cm^{-1} for Na-doped C60, and 500 S cm^{-1} for K-doped C60 film). (Nature, 1991, 350, 320) The K-doped C60 film displays metallic nature and becomes superconducting at T_c of 18 K. Note that the excellent conduction of alkali metal doped C60 films is most likely achieved at a much higher doping level than that of the doped PTEG-2 film: for example, the K-doped C60 film exhibits a low Seebeck coefficient of $-11 \mu\text{V K}^{-1}$ and C60 molecule is triple charged (a doping level of 3) upon K-doping (Phys. Rev. Lett. 1992, 69, 3797) while the doped PTEG-2 film has an Seebeck coefficient of $-223 \mu\text{V K}^{-1}$ and a doping level of ~ 0.12 . Additionally, the doped PTEG-2 film with n-DMBI is more conductive than previously reported co-evaporated C60/n-DMBI blend film (with an electrical conductivity of 5.5 S cm^{-1}) (JACS, 2012, 134, 3999). These results suggest that meticulously designed side chains could not only provide the fullerene derivative with much improved solution processability but also enable a comparable charge transport with that of C60 fullerene upon doping.

By following an empirical L-S relation for non-degenerate semiconductors reported by Snyder et al (APL Mater., 2015, 3, 041506), we estimated the Lorenz number of $L=1.65 \times 10^{-8} \text{ W}\Omega\text{K}^{-2}$ for the doped PTEG-2 film. Then the electronic contribution to the thermal conductivity (K_E) is calculated to be $0.004 \text{ Wm}^{-1}\text{K}^{-1}$, which is very small compared to the total thermal conductivity K . Therefore, we agree with the reviewer that the electrical conductivity is too low to impact the thermal conductivity of electrons K_E and the total K (K_E+K_L) originates from the lattice thermal conductivity (K_L).

We have changed the manuscript accordingly.

Reviewer #2:

The manuscript entails an experimental study of the thermoelectric performance of a molecularly doped fullerene derivative. A branched ethylene glycol side chain is found to result in a, for fullerenes, record electrical conductivity while preserving a low thermal conductivity. Overall, a for organic materials very high thermoelectric figure of merit is obtained; the value is among the highest values reported to date. The thermoelectric data are certainly very good, and should be published in a journal such as Nature Communications. However, the analysis of the data both in terms of (heat and electrical) transport physics and physical chemistry needs improvement. Therefore, I cannot recommend publication of the manuscript in its current form. I am sorry for not being more positive, but hope that my comments below will help the authors to further sharpen their manuscript.

1) My major concern is the proposition that the presented material is a phonon-glass electron-crystal. While this argument is feasible, the manuscript lacks proof that could substantiate such a claim. As the authors state, this insight would be a considerable advance for the field of organic semiconductors. At a minimum, the authors should present temperature dependent thermal and electrical conductivity data. That should allow to identify phonon-glass electron-crystal type behavior.

Response:

We thank the reviewer for stating that the thermoelectric data should be published in a journal such as Nature Communications.

The temperature dependent thermal conductivity and electrical conductivity can be found in Fig. 5A and 5C. Those data show that the thermal conductivity hardly depends on temperature, while the electrical conductivity shows a maximum around 120°C.

After carefully considering the reviewer's point, we drop the claim that we fully realized an organic 'PGEC' in the present study, and instead claim that we are close to it.

Unfortunately, there is no clear, strict definition of an organic PGEC in the current literature. Even for an inorganic PGEC it is difficult to find a proper definition. Therefore, in the revised version of our manuscript, we propose the following definition of an organic PGEC:

- *the thermal conductivity reaches the amorphous limit (<https://journals.aps.org/prb/abstract/10.1103/PhysRevB.46.6131>) of this material;*
- *the charge carrier mobility should reach the crystalline limit of this particular material.*

In this manuscript we show that the thermal conductivity does indeed reach (it's actually even lower than) the amorphous limit. For the related fullerene derivative PCBM there are several reports in the literature that show that the pristine material (i.e. undoped) does indeed satisfy this part of the definition. Here we show that 1) PTEG-2 has a similarly low thermal conductivity, and 2) it is not affected by doping. In sum, doped PTEG-2 is a phonon glass.

Regarding the charge carrier mobility: Frankevich et al. (Frankevich, E.; Maruyama, Y.; Ogata, H. Chem. Phys. Lett. 1993, 214, 39.) have found a (time-of-flight) mobility of $0.5 \pm 0.2 \text{ cm}^2/\text{Vs}$ for single-crystal C60 and Bao et al. (J. Am. Chem. Soc. 2012, 134, 2760–2765) reported a high field-effect transistor mobility of $5.2 \pm 2.1 \text{ cm}^2/\text{Vs}$ for a needle-like single crystal. It stands to reason that the mobility of C60 is an upper limit to that of fullerene derivatives as the side-chains will dilute the conjugated part of the system. In our manuscript, we find a mobility of $1.2 \text{ cm}^2/\text{Vs}$ which is of the same order of magnitude as that of single crystal C60. In other words, PTEG-2 comes close to (if not actually fulfills) the mobility requirement.

We note that the conductivity (see Fig 5a) shows little temperature dependence. Above 120°C the conductivity even shows a negative temperature dependence. Figure 5b (Seebeck coefficient) shows that the Seebeck coefficient does not change with temperature, suggesting that the number of charge carriers remains constant. Taken together, this suggests that the conductivity might even be metallic in the sense that it shows negative temperature dependence. However, we do not wish to claim that this is the case as that would require a more detailed study.

These properties indicate that the doped PTEG-2 film is very close to an organic 'PGEC' material. The corresponding changes have been made and marked in red in the main text.

2) the authors mention the doping efficiency repeatedly. I recommend that the authors define this term more carefully. Do the authors refer to a charge transfer event? Not each generated polaron will lead to a free charge. Ionization and dissociation of charges should be discussed.

Response:

We agree with the reviewer that not every generated polaron will lead to a free charge. The doping efficiency is defined as the free charge density to the number of introduced dopant molecules (marked in red on page 14 in the main text). We directly determined the free charge density of doped PTEG-2 films (annealed at different temperatures) by using Mott-Schottky analysis on ion-gel based metal-insulator-semiconductor devices, which is used to estimate the doping efficiency (47% for annealed at 120 °C and 60% for annealed at 150 °C). Based on the reviewer's comment, we added the electron paramagnetic resonance spectra of the pristine and doped PTEG-2 films annealed at different temperatures (Figure S7c). We find that 40% more polarons are generated upon annealing at 150°C instead of 120°C. The annealing temperature does not seem to strongly influence the dissociation probability of polarons into free charges. Therefore, more polarons lead to more free charges and, hence, increased doping efficiency. These results are consistent with the improved mixing between the host and the dopant molecules as shown by the morphology study.

3) in the introduction the authors write that organic materials are not toxic. That is possibly true for the semiconductors. However, molecules like DMBI are highly reactive. The SDS provided by Sigma states: acute toxicity. I suggest that the authors reconsider their claim.

Response:

We thank the reviewer for pointing out the toxicity of the dopant. We changed the 'non-toxic' into 'abundant' in the introduction part.

4) line 121. The authors speculate that the fullerene material changes its "arrangement" upon annealing. The authors should provide evidence that no solvent is trapped in the films. Even chloroform may be trapped, despite a low boiling point, which could explain the observed decrease in film thickness.

Response:

Yes, it is very important to exclude any solvent trapping for dynamic ellipsometry measurement. We took care to do so when preparing the samples. After spin-coating various fullerene derivative based films, we kept the samples in high vacuum ($<10^{-6}$ mbar) for two days in order to remove any trapped solvent. This strategy has proven very effective to remove high-boiling-point additives in polymer solar cells, for example (J. Phys. Chem. C 2013, 117, 14920–14928). Additionally, we performed the same measurement for vacuum-treated pristine PTEG-2 film pre-annealed at 90 °C (higher than the boiling point of chloroform) for 1 h and the result is displayed in Figure S1f. the d-T plot for this pristine PTEG-2 film shows the same feature: inflection point at $T=131$ °C, and the film shrinks in the 131 °C $< T < 155$ °C

range. Regarding this, we believe that the variation in thickness is attributed to the arrangement of fullerene derivatives rather than the removal of any trapped solvent molecules.

5) The film thickness during annealing is extracted from ellipsometry measurements. Detailed information about the model should be provided that was used to convert the ellipsometric angles into film thickness. How were the raw data fitted and how was the accuracy of the fit determined? At minimum, a least square analysis of the fit should be provided. And what range of wavelengths were fitted?

Response:

More detailed information about the model and the fitting were added in the Supplementary Information. The fitting parameters and the mean squared error (MSE) of fits are summarized in Table S1. We used the model, that is silicon substrate/silicon oxide/Cauchy layer, for fitting the spectroscopic data of all the fullerene derivative based samples. The thickness of the native oxide layer was determined by fitting the ellipsometry data of a blank silicon substrate with a native oxide layer. The Cauchy dispersion function ($n(\lambda)=A_n+B_n/\lambda+C_n/\lambda$, $k=0$, where n is the refractive index, k extinction coefficient, A_n , B_n and C_n are Cauchy parameters) was used to fit the optical property of the fullerene derivative based layers at various wavelengths. The ellipsometry data were fitted in the wavelength range from 700 nm to 1700 nm, in which the fullerene derivative based films show no light absorption. Figures S1a-e shows the experimental data and the corresponding model fits, which show very good overlap with each other. From the Figure S1a-e and the small MSE values, we think the experimental data were well fitted with a high accuracy.

6) The authors conclude from their measurements that the fullerene materials undergo a phase transition, and propose that side chain melting occurs. Line 171: “ T_{inf} can be physically interpreted as the melting point of the side chains”. I am not quite sure what a “side chain melting” is supposed to mean. In any case, a melting event would lead to a sudden decrease in density, and therefore an increase in thickness. The authors observe a decrease in thickness though. In my opinion, the suggested thermal behavior is pure speculation and must be substantiated with other types of measurements.

Response:

We thank the reviewer for pointing out this. We tried to express that at T_{inf} the side-chains become so flexible that the molecular arrangement of entire PTEG-2 molecules can change, which leads to a more densely packed film. We were, however, not suggesting that the entire compound would melt, which would indeed lead to a sudden decrease in density. However, a proper understanding of the exact roles of the side-chains is beyond the scope of this manuscript and we have removed the statements relating to the melting of the side-chains from the main text.

7) related to 6), the comment on line 174: “melting...provide[s] energy for rearrangement” makes no sense. How does melting provide energy? Do the authors mean the heat that is released by a melting event? Please measure that heat.

Response:

We measured the DSC for various fullerene derivative powder samples. Unfortunately, we did not observe apparent melting behavior of the side chains as reported by a previous study on a glycolated polythiophene (Li et al. Org. Electron. 2016, 33, 23). One possible reason might be the low weight percentage of the side chain in fullerene derivatives, as compared to that in the glycolated polythiophene.

So, we deleted the part “Linear ethylene-glycol-type side chains of three units have been reported to melt at 98 °C.³⁵ The side chains of PPEG-1 and PTEG-2 are longer or more bulky and should therefore melt above 98 °C. The T_{inf} values of the fullerene derivatives vary with the side chains in the order PPEG-1 < PTEG-2 < F2A, which correlates well with the melting points of these side chain types. Thus, we surmise that T_{inf} can be physically interpreted as the melting point of the side chain of a fullerene derivative. This speculation is reasonable because the melting of a side chain should increase the spatial freedom of fullerene-based molecules and provide energy for rearrangement into more energetically favorable positions.”

8) line 171. The authors insinuate that the polarity of side chains improves the thermal stability. Only one of the polar side chains (those of PTEG-2) show good thermal stability. Surely, it is not the polarity but some other parameter that determines the stability. That aside, I agree with the authors that their observation of a thermally stable doped system is significant.

Response:

We thank the reviewer for pointing out this. We measured the thermal stability of the doped F2A with the alkyl side chain under the same condition (at 150 °C) and the result is displayed in Figure 2b together with those for other fullerene derivatives (PTEG-1, PPEG-1, and PTEG-2). Of these materials, the ones with polar side chains are more thermally stable than the one with an alkyl side chain (F2A), which agrees with previous reports. Therefore, the polarity of the side chain appears to play some role in the thermal stability of the doped films. We do not exclude the effect of other parameters like the geometry of the side chain. We have changed the text accordingly.

9) line 200. “aggregates gradually disappear. How is the presence and disappearance of aggregates included in the ellipsometric model that was used to determine the film thickness. Did the authors assume a multilayer stack? If the model does not include at least a bilayer (the top layer represents the aggregates) that disappears upon annealing, then the observed changes in thickness could easily be an artefact. I must thus question the ellipsometry analysis.

Response:

From the dynamic ellipsometry measurement, we obtained a plot of the ellipsometry parameter (ψ) as a function of the temperature. The ellipsometry parameter $\psi(T)$ is related to contribution of the entire organic film, which is one layer for the un-doped film or bilayer for the doped one. As the organic systems do not undergo decomposition during the annealing process, the weight of the sample is constant, and thus the thickness variation corresponds to the volume change with the temperature, which inversely scales with the density variation of the film. Therefore, the thickness should be a parameter translated by the volume of the organic material excluding the void part. For this reason, we used one Cauchy layer to describe the organic film and extract the $d(T)$ plots for various pristine and doped fullerene derivatives in Figure 2a. We also used a bilayer model for the doped PTEG-2 film and the

bilayer includes a Cauchy layer and an atop composite layer consisting of the Cauchy material and void with a certain percentage (f). The thickness (d_2) of the composite layer was considered to be the height (30 nm-70 nm) of the aggregates measured by the AFM and is set to 50 nm in the model. By fitting the experimental data, the thickness (d_1) of the Cauchy layer and the percentage of the void were determined to be 89.7 nm and 96.4%, respectively. Therefore, the volume-translated thickness (d) of organic material could be expressed by: $d=d_2+d_1 \cdot (1-f)$ and we obtain $d=91.5$ nm, which is very close to $d=91.6$ fitted by the one layer model. Therefore, we think the one layer model provides a good approximation for the volume-translated thickness for the doped PTEG-2. In addition, the valley in the $d(T)$ curve for the doped PTEG-2 film was also observed in the $d(T)$ plot for the pristine PTEG-2 film that has no aggregates on the surface. As such, we believe the inflection point in the $d(T)$ plot corresponds to the phase transition of the organic material.

10) Figure 3 and paragraph starting at line 235. The MD simulations are very illustrative. I would like to see a simulation of the diffraction pattern in Fig 3b, which substantiates the proposed unit cell.

Response:

We computed (procedure in the methods and SI of the revised manuscript) the diffraction patterns for the MD unit cells along the q_z and q_y directions which can be compare to the experimental ones- see updated Figures 3c and 3d:

The agreement with the experimental scattering line cuts (Fig 3c and 3d, respectively) is very good. We note here that the calculated scattering curves come from averaging over 240 perfectly oriented 100-nm sized crystals, while this is not the case for the experimental scattering signals, where some “misaligned” crystals can: 1) broaden the peaks; 2) introduce some of the peaks which would belong to the q_z direction into the q_y scattering signal and vice versa. We think that this is the reason why, for example, the peak observed in the low- q region along the q_y (Fig 3d.) is not present in the simulated scattering pattern, that peak being likely coming from a misaligned (i.e., rotated by 90 degree) PTEG-2 crystallite.

11) line 249. Now the ethylene glycol side chains are described as having different configurations. How is this consistent with the picture provided earlier where these side chains order at low temperatures, and upon heating?

Response:

We acknowledge that the description in the text was perhaps misleading and we rephrased now the text to (previous line 245-250):

'A common feature of the multiple configurations obtained from the MD simulations is a staggered arrangement for the C₆₀ bilayers interposed by the ethylene glycol phase. However, the MD simulations do not converge all into one specific configuration of the ethylene glycol chains, but rather give an ensemble of similar ones. The convergence into a single ethylene glycol configuration is likely prevented by the fact that the ethylene glycol chains are quite flexible due to the low energy barriers between their different configurations.'

12) line 272. A doping efficiency of 60% is calculated. Are these polarons formed per dopant molecule or free charges? And how does this value compare to the proposed disappearance of aggregates (line 200), which leads to a higher conductivity. Does the doping efficiency increase upon annealing? The authors quote carrier densities at 120 and 150 deg but only one doping efficiency. Is this because the doping efficiency does not actually increase, which would be inconsistent with the earlier argument that more dopant is taken up by the fullerene material?

Response:

The doping efficiency is defined by the ratio of the free charge carrier density to the number of the dopant molecules. We used the ion-gel based metal-insulator-semiconductor (MIS) devices along with Mott-Schottky analysis to extract the free charge carrier density. The effectiveness of this approach has been proven in our previous work (Adv. Mater. 2018, 30, 1704630). The doping efficiency of doped PTEG-2 film is calculated to be around 47% after annealing at 120 °C and 60% after annealing at 150 °C. The increased doping efficiency at the higher annealing temperature is likely to result from the improved mixing between the host and the dopant molecules. This was further substantiated by the EPR measurement that indicates more polarons are generated upon annealing at 150 °C (Figure S7c). We changed the text accordingly.

13) line 298. Now the side chain melting occurs at 120 deg. On line 169 it is 98 deg.

Response:

In the original version of the manuscript, On line 169, 98 °C is the melting point of the linear triethylene glycol type side chain, reported in the Ref. 35 (Li et al. Org. Electron. 2016, 33, 23). On line 298, 120 °C is assumed to be the melting point of double triethylene glycol type side chain, which is more bulky than the linear triethylene glycol type side chain. However, as the reviewer pointed out, we lack sufficient evidence to support this point. Therefore, we deleted the corresponding part discussing whether or not the melting of the side chain initiated the phase transition.

Minor comments:

1) line 70. Explain "proper n-doping"

Response:

We rephrasing 'proper n-doping' with 'Carbon nanotubes (CNTs) are able to exhibit $\sigma > 1000 \text{ S cm}^{-1}$ upon doping with n-type dopants such as benzyl viologen dichloride and crown ester complex salts (J. Mater. Chem. A, 2017, 5, 15631 and Adv. Funct. Mater. 2016, 26, 3021)'.

2) line 259 and 267. Provide error bars for the Seebeck coefficient and carrier densities.

Response:

The error margins for the Seebeck coefficient and the carrier densities are provided in the main text.

3) The doped fullerene material is not stable under ambient conditions. All electrical characterization was done under nitrogen. But how about the thermal conductivity measurement. How did the authors ensure that the sample had not degraded? The Linseis system requires that the sample is handled in air. Was the instrument placed in a glove box?

Response:

We agree with the reviewer that most n-doped organic semiconductors including doped fullerene materials are not stable under ambient conditions. Importantly, the main doping process of fullerene derivative/n-DMBI system does not occur directly by blending with a solution process and requires post-annealing above 90 °C to be activated. The ascast PTEG-2/n-DMBI blend (5 wt% n-DMBI) film exhibits an electrical conductivity of $2.6 \times 10^{-4} \text{ S cm}^{-1}$. We found that air exposure of as-cast fullerene derivative films has little influence on the eventual electrical conductivity if the as-cast fullerene derivatives are post-annealed in an inert environment or vacuum. This applies to most fullerene derivatives such as PCBM, PTEG-1, PPEG-1, and PTEG-2, which are doped by n-DMBI. Air exposure of 5 wt%-doped PTEG-2 film for 1 h causes a deviation of the eventual conductivity less than 10%. Therefore, before the thermal conductivity measurement, we transferred the as-cast PTEG-2/n-DMBI blend film into the Linseis setup and carried out the in situ thermal annealing at 150 °C in vacuum to complete the doping process. The thermal conductivity was also measured by the IIT group. They mounted the samples into an air-tight chamber with optical windows in a N₂-filled glovebox. Before thermal conductivity measurement, the air-tight sample chamber was pumped down with a vacuum of $<10^{-5}$ torr. There was no air exposure during the measurement.

Reviewer #3:

Liu et al. claimed that they developed OTE materials with ZT value over 0.3 following PGEC concept. In fact, this concept is widely used in inorganic materials, but its applications in OTE materials remain unclear. Although the basic concept seems interesting, I have the follow concerns about the results.

(1) For typical PGEC concept in inorganic materials, the designed atoms are introduced into crystal cell to ensure high electrical conductivity and suppress the lattice thermal conductivity. Although the authors claimed their designed materials is similar the concept, I failed to find a clear definition of the concept in organic materials. Notably, several doped organic semiconductors have been confirmed to possess ordered molecular packing and the dopant is located at side-chain regime. Once again it raises a question, how could define PGEC materials in conjugated systems? Whether these previous reported materials also follow the concept?

Response:

We agree with the reviewer that there is no clear definition of the PGEC concept as applied to organics. In the new version of our manuscript, we have attempted to introduce a suitable definition:

- *the thermal conductivity reaches the amorphous limit (<https://journals.aps.org/prb/abstract/10.1103/PhysRevB.46.6131>) of this material;*
- *the charge carrier mobility should reach the crystalline limit of this particular material.*

In this manuscript we show that the thermal conductivity does indeed reach (it's actually even lower than) the amorphous limit. For the related fullerene derivative PCBM there are several reports in the literature that show that the pristine material (i.e. undoped) does indeed satisfy this part of the definition. Here we show that 1) PTEG-2 has a similarly low thermal conductivity, and 2) it is not affected by doping. In sum, doped PTEG-2 is a phonon glass.

Regarding the charge carrier mobility: Frankevich et al. (Frankevich, E.; Maruyama, Y.; Ogata, H. Chem. Phys. Lett. 1993, 214, 39.) have found a (time-of-flight) mobility of $0.5 \pm 0.2 \text{ cm}^2/\text{Vs}$ for single-crystal C60 and Bao et al. (J. Am. Chem. Soc. 2012, 134, 2760–2765) reported a high field-effect transistor mobility of $5.2 \pm 2.1 \text{ cm}^2/\text{Vs}$ for a needle-like single crystal. It stands to reason that the mobility of C60 is an upper limit to that of fullerene derivatives as the side-chains will dilute the conjugated part of the system. In our manuscript, we find a mobility of $1.2 \text{ cm}^2/\text{Vs}$ which is on the same order of magnitude as that of single crystal C60. In other words, PTEG-2 comes close to (if not actually fulfills) the mobility requirement.

All in all, we have revised our claim by stating that our material system approaches the PGEC concept.

A few doped organic semiconductors have been confirmed to possess ordered molecular packing and the dopant is located at the side-chain regime. We are not sure whether or not they can be qualified as 'organic electron crystals' only on the basis of the microstructure, as a proper PGEC also has very good electrical transport properties (see definition above).

(2) The ultralow thermal conductivity and high ZT are the key evidences for the so-called PGEC concept. I have several concerns about measurement of the κ and ZT values:

i) As a well-known fact, it is challenge to measure sample with low in-plane thermal conductivity ($\kappa \leq 0.1 \text{ W/m K}$). Even with the technique provided in the supporting information, I'm not sure about the accuracy of results. A preferred thickness of the sample should be $> 400 \text{ nm}$ for a sample with κ of 0.4 W/m K (Journal of Electronic Materials 2018, 47 (6) , 3203–3209. DOI: 10.1007/s11664-017-5989-4). I believe that the situation is very challenge for PTEG-2 film since much thicker film is required. The author should provide the exact thickness of the sample used in TFA measurement and evaluate the deviation.

Response:

Yes, it is really a challenge to measure the thermal conductivity of the fullerene derivative materials. We measured the thermal conductivity of the pristine and doped PTEG-2 films which are several micrometers thick, for several times and similar results are obtained. For the result shown in Figure 5c, the thickness is $6.71 \pm 0.23 \mu\text{m}$ for the doped PTEG-2 film and $3.71 \pm 0.35 \mu\text{m}$ for the pristine PTEG-2 film. The obtained thermal conductivity values are in agreement with the reported (by two different groups) literature values for PCBM ($0.03 \text{ Wm}^{-1}\text{K}^{-1}$ Phys. Rev. Lett. 2013, 110, 015902 and $0.06 \text{ Wm}^{-1}\text{K}^{-1}$ Phys. Rev. B. 2013, 88, 075310), which supports our findings.

ii) I note the measurement of σ and S were performed in a N_2 -controlled environment while κ_{\parallel} was obtained in vacuum by TFA. What about the air stability of the sample during the sample transfer? Such kind of n-type materials usually possess poor air stability, which will lead to rapid de-doping and lower κ_{\parallel} because of reduced contribution from electron transport. In fact, TFA can characterize all the parameters and even temperature dependent performance in one chip. Why the author prefers to utilize different techniques to characterize the samples?

Response:

We agree with the reviewer that moderately/heavily doped organic semiconductors are not stable under ambient conditions. The main doping process within fullerene derivative/n-DMBI films does not occur directly after the spin-coating process and requires post-annealing above $90 \text{ }^\circ\text{C}$ to be activated. The as-cast PTEG-2/n-DMBI blend (5 wt% n-DMBI) film exhibits an electrical conductivity of $2.6 \times 10^{-4} \text{ S cm}^{-1}$. We found that air exposure of as-cast fullerene derivative films has little influence on the eventual electrical conductivity if the as-cast fullerene derivative films are post-annealed in an inert environment or vacuum. This applies to most fullerene derivatives such as PCBM, PTEG-1, PPEG-1, and PTEG-2, which are doped by n-DMBI. Air exposure of as-cast 5 wt%-doped PTEG-2 film for 1 h causes a less than 10% deviation of the eventual conductivity upon annealing at $150 \text{ }^\circ\text{C}$ for 1 h. Therefore, before thermal conductivity measurement, we transferred the as-cast PTEG-2/n-DMBI blend film into the Linseis TFA setup and carried out the thermal annealing at $150 \text{ }^\circ\text{C}$ in vacuum to complete the doping process. In this way, we avoid to underestimate thermal conductivity because of the electronic contribution loss. The thermal conductivity was also measured by the IIT group. They mounted the samples into an air-tight chamber with optical windows in the N_2 -filled glovebox. Before thermal conductivity measurement, the air-tight sample chamber was pumped down with a vacuum of $<10^{-5}$ torr. There was no air exposure during the measurement.

As for the thermoelectric characterization, we performed all the electrical conductivity and Seebeck coefficient measurement in the N_2 -filled glove-box in our lab in the University of Groningen. The data shown in this work have been reproduced for several times. Furthermore, we would like to keep this work consistent with our previous reports, where the same method was used to measure the electrical conductivity and Seebeck coefficient. However, we have no facility to measure the thermal conductivity of the organic film. Therefore, we collaborated with Prof. Derya Baran's group at KAUST for the thermal conductivity measurement of the pristine and doped PTEG-2. They are experts in studying thermal transport of thin film samples (Advanced Science, 2020, 1903389). In order to make sure the sample quality, they checked the electrical conductivity of the doped film, which is $>8 \text{ S cm}^{-1}$ very close to the value obtained at the University of Groningen. The measurement of the thermal conductivity has been repeated for several times and consistent results were obtained, which is displayed in the main text.

iii) It is difficult to understand that the σ shows a large variation versus temperature ($>30\%$) while the κ_{\parallel} almost keeps nearly unchanged in Figure 5. The electronic contribution of the κ is related to the σ and follows Wiedemann-Franz law. In other words, one should observe the temperature dependent κ

even for PGEC materials. The authors should identify the electronic contribution and lattice contribution to the κ to confirm and understand the phenomenon.

Response:

We would like to first estimate the electronic and lattice contribution to the thermal conductivity of the doped PTEG-2 film. By applying an empirical L-S relation for non-degenerate semiconductors reported by Snyder et al (APL Mater., 2015, 3, 041506) to this study, we estimated the Lorenz number of $L=1.65 \times 10^{-8} \text{ W}\Omega\text{K}^{-2}$ for the doped PTEG-2 film. Based on the Wiedemann-Franz law, the electronic contribution to the thermal conductivity (K_E) is calculated to be $0.004 \text{ Wm}^{-1}\text{K}^{-1}$, which is very small compared to the total thermal conductivity K . The thermal conductivity of the doped PTEG-2 film is dominated by the lattice contribution.

3) The authors found that the electrical conductivity first rises then falls along with the increase temperature. They attribute the decreased σ to the enhanced disorder of the side chain. Providing the energetic disorder before and after phase transition will make it clearer. In fact, dedoping can also contribute to the decrease of the conductivity. What about the thermal stability of doped films?

Response:

The doped PTEG-2 film is very stable under the thermal stress as shown in Figure 2b. Actually, after cooling the sample down to room temperature, we measured the electrical conductivity of $\sigma=8.0 \text{ S cm}^{-1}$ at 25°C (the red star in Figure 5a), which is very close to its original value. It indicates that the decrease of electrical conductivity with temperature above 120°C is a reversible process and was not caused by the de-doping of the sample. We speculate that, at higher temperatures, molecular vibrations hamper charge transport, somewhat similar to phonon scattering in an inorganic semiconductor. We made corresponding changes in the main text, which was marked in red.

4) The doped PTEG-2 film shows high crystallinity. Since the author use ‘electron crystal’ to describe the materials, I suggest the authors to evaluate of the carrier concentration and mobility by TFA to support the definition.

Response:

Unfortunately, the Linseis TFA setup at KAUST does not support the measurement of the carrier density and mobility of thin film samples. Instead, we measured the free charge density of the doped PTEG-2 films by using admittance spectroscopy on ion-gel-based metal insulator semiconductor (MIS) device, which has previously proved effective for direct measurement of charge density for the moderately or heavily doped organic semiconductors (Adv. Mater. 2018, 30,1704630). We determined the free charge density of $(4.5 \pm 0.3) \times 10^{19} \text{ cm}^{-3}$ and bulk mobility of $\sim 1.2 \text{ cm}^2 \text{ V}^{-1} \text{ s}^{-1}$ for the doped PTEG-2 upon annealing at 150°C . As argued in the above (see definition of PGEC) this is on par with crystalline C60.

Reviewers' Comments:

Reviewer #1:

Remarks to the Author:

The authors have made appropriate revisions and the manuscript is acceptable for publication in my opinion.

Reviewer #2:

Remarks to the Author:

All three reviewers have commented on the lack of proof that the material studied in this manuscript is a "phonon-glass electron-crystal" as already suggested in the title. The authors use reference 7 to loosely define such a PGEC as a material with: "(i) the thermal conductivity reaches the amorphous limit of the particular material, and (ii) the charge carrier mobility should reach its crystalline limit."

However, if reference 7 is read carefully, these conditions do not describe the definition of a PGEC but its resulting properties, which are not exclusive to a PGEC. A PGEC is defined as a particular crystal structure. According to reference 7, a "PGEC material features cages or tunnels in its crystal structure inside which reside massive atoms that are small enough relative to the cage to "rattle"." The authors provide zero proof that their fullerene material has such a crystal structure.

In fact, reference 7 states: "To date, it has not been proven that a PGEC material exists. In most cases of purported PGEC materials, other factors coexist which also account for the observed reduction in thermal conductivity." I think it is bold to now transfer the PGEC terminology to organic materials, without any proof, hoping that the term "will stick" with the community. It just spreads confusion.

The authors write that they present a material "that approaches an organic 'PGEC' TE material." This essentially means that the material is not a PGEC.

I strongly encourage the authors to drop the PGEC formalism from their manuscript, and instead focus on a fact-based description of the properties of their materials. Nature Communications encourages a discussion section at the end of a manuscript. This would be a perfect place to point out that the presented material shares many properties reminiscent of a PGEC, and that further research may in fact show that such materials can exist among organic semiconductors.

Minor comment. Please plot the MSE as a function of temperature for Figure 2a. Changes in thickness can arise because of a sudden change in MSE, and then reflect a limit in the fitting accuracy rather than a physical change such as a phase transition.

Reviewer #3:

Remarks to the Author:

The authors have well addressed my concerns by performing additional experiments. Moreover, they also changed the claim with regard to the "phonon-glass electron-crystal" concept. I recommend its publication in the present form.

General remarks

We thank the Reviewers for their favourable assessment of our manuscript. Reviewers 1 & 3 recommend the manuscript be published. Below, we address the concerns raised by Reviewer 2. We have highlighted the changes made to the text of the manuscript.

Reviewer #1

The authors have made appropriate revisions and the manuscript is acceptable for publication in my opinion.

Reviewer #2

All three reviewers have commented on the lack of proof that the material studied in this manuscript is a “phonon-glass electron-crystal” as already suggested in the title.

All reviewers did indeed comment on the PGEC concept as discussed in the original version of the manuscript. We used the remarks by all Reviewers to significantly improve the original version and, as a result, Reviewers 1 and 3 recommend the manuscript be published without further changes. However, in doing so, we used a rather unfortunate wording of the PGEC concept on page 3 (introduction): it looked like the PGEC concept is based on Ref 7, which it is not. In the current version of the manuscript we have sought to rectify this mistake (see below) and to remove any confusion about the original definition, our interpretation, and our claim.

The authors use reference 7 to loosely define such a PGEC as a material with: “(i) the thermal conductivity reaches the amorphous limit of the particular material, and (ii) the charge carrier mobility should reach its crystalline limit.”

There might be some confusion about the origin of the PGEC concept, possibly as a result of our wording (which we have changed). We did not use ref. 7 to define the PGEC concept but rather reference 4, the original work by Slack. Possible realisations of this concept appear in reference 7 (and in many other papers). In Ref 4 (page 411), Slack writes:

‘In surveying various known candidates for desirable properties a material is required that resembles a phonon glass and an electron single crystal or “PGEC”. This means a material in which the phonon mean free paths are as short as possible and in which the electron mean free paths are as long as possible.’

Our definition (or interpretation) of the PGEC concept is based on Slack’s original wording, not on Ref. 7. This source (Ref. 4) is not so easily accessible as it is only featured in the 1995 edition of the CRC Handbook of Thermoelectrics. For the Reviewer’s convenience, we have included a pdf scan of the relevant part (p. 411) of Ref. 4 (see next page).

In order to make it possible to determine whether a material is a PGEC, we have rephrased Slack’s definition in terms of physical quantities that are easily experimentally accessible (i.e. thermal conductivity and charge carrier mobility).

FIGURE 3 The lattice thermal conductivity of PbTe and a mixed crystal of $\text{PbTe}_{0.5}\text{Se}_{0.5}$ as a function of temperature. The curve labeled λ_{min} has been calculated for PbTe if all phonons have a mean free path equal to one wavelength.

such materials are expected to be elements or binary or ternary compounds formed from the heavy elements near the bottom of the periodic table. One example is elemental bismuth and bismuth-antimony mixed crystals, whose use in thermoelectric coolers has been reviewed by Yim and Amith.¹² As can be seen in Figure 2, these are not adequate to meet the required ZT goal. Elemental Te, even though it is a semiconductor, is not adequate. Its mobility is too low and its thermal conductivity is too high. Furthermore, it has a chain-like structure and only conducts electricity well along the z-axis. Thus, compounds should be used instead of elemental semiconductors as candidate materials. This means that primarily compounds of Sn, Sb, Bi, or Te with various heavy metals are of interest.

PGEC

In surveying various known candidates for desirable properties a material is required that resembles a phonon glass and an electron single crystal or "PGEC". This means a material in which the phonon mean free paths are as short as possible and in which the electron mean free paths are as long as possible. Consider PbTe as a starting point. The thermal conductivity, λ , of pure PbTe vs. temperature is shown in Figure 3, and is based on data from the literature.²¹⁻²⁴ λ vs. T for a mixed crystal of 50 mol% PbTe plus 50 mol% PbSe given in Figure 3 is based on other measurements.²⁵⁻²⁹ The data from 28 to 90 K is from Soltys.²⁶ There is another data point for $\text{Pb}_{0.5}\text{Ag}_{0.25}\text{Sb}_{0.25}\text{Te}$ from Rosi et al.,³⁰ who found a thermal conductivity at room temperature of

of temper-
low is along

3, one can
means "a
: possess a
values. We
e within a
1 between

free paths
minimum
'stals have
 λ_{min} . The
ong-range
non mean

eater than
| by Ioffe¹

However, if reference 7 is read carefully, these conditions do not describe the definition of a PGEC but its resulting properties, which are not exclusive to a PGEC. A PGEC is defined as a particular crystal structure. According to reference 7, a “PGEC material features cages or tunnels in its crystal structure inside which reside massive atoms that are small enough relative to the cage to “rattle”.” The authors provide zero proof that their fullerene material has such a crystal structure.

It is actually the exact opposite. Reference 7 illustrates how nano-structuring is a possible, worthwhile strategy for realizing the PGEC concept. It is not, however, the proper definition of the concept itself. The PGEC concept is not defined in terms of a crystal structure but through its properties (see above).

In fact, reference 7 states: “To date, it has not been proven that a PGEC material exists. In most cases of purported PGEC materials, other factors coexist which also account for the observed reduction in thermal conductivity.” I think it is bold to now transfer the PGEC terminology to organic materials, without any proof, hoping that the term “will stick” with the community. It just spreads confusion.

We fully agree with the Reviewer that confusion should be avoided. That is why we have revised our wording where the PGEC concept is introduced (page 3, Introduction) and have made the reference to Slack’s work more precise (i.e. we specify the exact page number in the reference list). We also explain how Slack’s definition is similar to what we proposed in our work (page 3, bottom).

Regarding reference 7, please note that we did not aim to reduce the lattice thermal conductivity as it is known to be very low for a similar fullerene derivative (PCBM), although we did verify this experimentally. Our material, just like PCBM, behaves like a ‘Phonon Glass’. Rather, we sought to improve the electron transport and bring it up to the value of single crystals, thus also fulfilling the ‘Electron Crystal’ part of the PGEC.

The authors write that they present a material “that approaches an organic ‘PGEC’ TE material.” This essentially means that the material is not a PGEC.

The high bulk mobility ($> 1 \text{ cm}^2/\text{Vs}$) of our system is of the same order of magnitude as that of single crystal C60, indicating that doped PTEG-2 comes close to the mobility requirement of an organic ‘electron crystal’. We say ‘comes close’ just to err on the side of caution. The very fact that we get such high mobilities is one of the key results of the work and drives the (record-)high ZT-value. To make it more clear that we do not claim to have reached a PGEC material (i.e. approached), we have removed PGEC from the title.

I strongly encourage the authors to drop the PGEC formalism from their manuscript, and instead focus on a fact-based description of the properties of their materials. Nature Communications encourages a discussion section at the end of a manuscript. This would be a perfect place to point out that the presented material shares many properties reminiscent of a PGEC, and that further research may in fact show that such materials can exist among organic semiconductors.

We have modified the title (removed PGEC) and have changed the description of the original PGEC concept and how we apply it to organic semiconductors.

Minor comment. Please plot the MSE as a function of temperature for Figure 2a. Changes in thickness can arise because of a sudden change in MSE, and then reflect a limit in the fitting accuracy rather than a physical change such as a phase transition.

There might be a slight misunderstanding as to how the data in Fig. 2a were obtained (see the Supporting Information for details). We use a well-established method (Adv. Funct. Mater. 2014, 24, 2116; J. Mater. Chem., 2011, 21, 10676), which consists of

- 1) obtaining the thickness at room temperature (RT) by doing a least-squares fit to the ellipsometry data of the films measured at room temperature.
- 2) next, the ellipsometry parameter at different temperatures ($\psi(T)$) (at a wavelength of 800 nm and incident angle of 70°) was obtained by dynamic spectroscopy (temperature ramp scan with a rate of $2.5^\circ\text{C}/\text{minute}$ and a range of 25°C to 200°C).
- 3) The thickness at elevated temperatures was obtained by using a previously established relation between ψ and the thickness (Adv. Funct. Mater. 2014, 24, 2116).

Accordingly, with the thickness at RT and $\psi(T)$, we obtained the thickness as a function of the temperature ($d(T)$). We cannot, therefore, plot the MSE as a function of temperature based on this, established, measurement technique. Moreover, the extracted thicknesses are thus not affected by changing in the quality of the fits with temperature.

A spectroscopic scan requires 10 minutes from 300 nm to 1700 nm at each temperature, and the films may change during the measurement which should be avoided. Therefore, we did not perform the spectroscopic scan at each temperature to obtain the thicknesses by fitting. However, in order to address the concerns of the reviewer, we have performed a complementary experiment: performing the spectroscopic scan at different temperatures ranging from RT to 180°C and least-squares fits were carried out for each spectroscopic plot for the doped PTEG-2 film. We use the following expression for the MSE

$$\text{MSE} = \sqrt{\frac{1}{2N-M} \sum_{i=1}^N \left[\left(\frac{\Psi_i^{\text{exp}} - \Psi_i^{\text{mod}}}{\sigma_{\Psi,i}^{\text{exp}}} \right)^2 + \left(\frac{\Delta_i^{\text{exp}} - \Delta_i^{\text{mod}}}{\sigma_{\Delta,i}^{\text{exp}}} \right)^2 \right]}$$

where N is the number of measured wavelengths, M is the number of fit parameters in the model, and the σ 's are the standard deviations of the experimental data points at each wavelength. As shown in the Figure below, the MSE does not show large changes over temperature variations and is always below 0.8, so the error in fitting is smaller than the experimental error at each temperature.

Reviewer #3

The authors have well addressed my concerns by performing additional experiments. Moreover, they also changed the claim with regard to the "phonon-glass electron-crystal" concept. I recommend its publication in the present form.

Reviewers' Comments:

Reviewer #2:

Remarks to the Author:

The authors have addressed all my comments and I recommend publication without change. I congratulate the authors to their results, which I am sure will be well received by the organic thermoelectrics community.